# Computational Analysis of Triazole-Based Kojic Acid Analogs as Tyrosinase Inhibitors by Molecular Dynamics and Free Energy Calculations

**DOI:** 10.3390/molecules27238141

**Published:** 2022-11-23

**Authors:** Lucas Sousa Martins, Reinaldo W. A. Gonçalves, Joana J. S. Moraes, Cláudio Nahum Alves, José Rogério A. Silva

**Affiliations:** 1Laboratório de Planejamento e Desenvolvimento de Fármacos, Instituto de Ciências Exatas e Naturais, Universidade Federal do Pará, Belém 66075-110, Brazil; 2Programa de Pós-Graduação em Química Medicinal e Modelagem Molecular, Instituto de Ciências da Saúde, Universidade Federal do Pará, Belém 66075-110, Brazil

**Keywords:** MD simulations, binding free energy, LIE, kojic acid analog, triazoles, tyrosinase

## Abstract

Molecular docking, molecular dynamics (MD) simulations and the linear interaction energy (LIE) method were used here to predict binding modes and free energy for a set of 1,2,3-triazole-based KA analogs as potent inhibitors of Tyrosinase (TYR), a key metalloenzyme of the melanogenesis process. Initially, molecular docking calculations satisfactorily predicted the binding mode of evaluated KA analogs, where the KA part overlays the crystal conformation of the KA inhibitor into the catalytic site of TYR. The MD simulations were followed by the LIE method, which reproduced the experimental binding free energies for KA analogs with an *r*^2^ equal to 0.97, suggesting the robustness of our theoretical model. Moreover, the van der Waals contributions performed by some residues such as Phe197, Pro201, Arg209, Met215 and Val218 are responsible for the binding recognition of 1,2,3-triazole-based KA analogs in TYR catalytic site. Finally, our calculations provide suitable validation of the combination of molecular docking, MD, and LIE approaches as a powerful tool in the structure-based drug design of new and potent TYR inhibitors.

## 1. Introduction

Melanin is a general name for a class of natural pigments derived from tyrosine that is found in many species of living organisms and microorganisms such as plants, animals, bacteria and fungi. It has many key functions including thermoregulation, photoprotection and healing process [1]. In plants, it is related to the browning of vegetables and fruits, which is an important aspect of the agriculture industry [2]. In mammals, melanin is crucial for the protection of skin and eyes against UV light [3]. Due to playing key roles in cell protection, its abnormal production is related to several hyperpigmentation disorders, such as senile lentigines, freckles and melasma [4], which provides evidence for the development of skin-whitening and depigmenting compounds based on melanin inhibition, which is important to the cosmetic industry [5]. Moreover, there is some evidence relating to neuromelanin and Parkinson’s disease [6].

The biosynthetic path of melanin involves several steps including the hydroxylation of _L_-Tyrosine (_L_-Tyr) to _L_-3,4-dihydroxyphenylalanine (_L_-DOPA), a monophenolase stage (1), followed by the subsequent oxidation of _L_-DOPA to *_O_*-dopaquinone, a diphenolase stage (2) [7] (Figure 1). Both reactions are catalyzed by a type-3 copper metalloenzyme called Tyrosinase (TYR), which contains a binuclear copper active site in which each Cu^2+^ ion chelates with three histidine (His) amino acid residues [8].

Despite the critical role of TYR in the melanogenesis and browning process, several studies have involved the design, synthesis and biological evaluation of TYR inhibitors [9,10], among which arbutin [11], hydroquinone [12], azelaic acid [13] and kojic acid (KA) [14] can be highlighted. Particularly, KA is used as a positive control for TYR inhibition, but it has shown side effects related to a high-sensitizing potential and considerable toxicity [15,16], which compromise its use in the cosmetical and pharmaceutical industries. Moreover, studies highlight the urgency for the development of new TYR inhibitors [7,10]. Recently, Ashooriha et al. [17] synthesized a set of KA analogs with high anti-TYR activity. They applied a click chemistry reaction and the formation of a 1,2,3-triazole ring to synthesize these new and potent TYR inhibitors [17]. Particularly, the most potent compounds (**6o** and **6p**) show lower cytotoxic activity than KA. Furthermore, the synthesized compounds can be added to the group of products of click chemistry application related to the Nobel Prize in Chemistry 2022.

Here, we reported a powerful computational analysis of 1,2,3-triazole-based KA analogs, synthesized and evaluated by Ashooriha et al. [17], by applying molecular docking, molecular dynamics (MD) simulations and binding free energy calculations. This study shines a light on the TYR inhibition mechanism by providing structural and energetic information that agrees with experimental proposals. Moreover, all applied computational procedures here have been successfully validated by our research group [18,19,20,21].

## 2. Results and Discussion

### 2.1. Molecular Docking and MD Simulations 

Initially, it should be highlighted that our molecular docking results using the Virtual Docker (MVD) package and MOLDOCK method [22] have been successfully applied for TYR systems [18,19,20,21]. In Figure 2, the docked conformation of the weakest and strongest potent 1,2,3-triazole-based KA inhibitor (**6h** and **6o**, respectively) show suitable conformations into the TYR catalytic site where the KA ring is close to the crystallographic structure of the KA complex. The main difference between the two structures is the orientation of the 1,2,3-triazole part of the KA analog, which also could not be resolved from the experimental electron density and is exposed to solvent. From the docking calculations, a MOLDOCK scoring function for each KA analog could be extracted and compared to experimental binding data (Table 1). There is no correlation (*r*^2^ = −0.23) between MOLDOCK scoring and the experimental data (IC_50_, µM) (see Appendix A). These results are not a surprise for molecular docking calculations, where docking algorithms in many cases can provide a suitable binding mode but cannot rank different ligands by affinity [23].

In particular, the hydroxyl groups of the KA part of 1,2,3-triazole-based KA inhibitors (**6a**–**6p**) interact with Cu^2+^(B) (Figure 2), producing distances of about 3.30 Å. Furthermore, the KA ring of 1,2,3-triazole compounds is involved in a hydrophobic pocket created by Met215, Gly216, Val217, Val218 and Ala224, as found in the KA inhibitor. Among the most important interactions, we can highlight the interaction with a key residue Arg209 through the π–cation stacking interaction with 1,2,3-triazole ring present in all KA analogs, for the natural substrates, this residue is responsible to form hydrogen bonds with the carboxylic group of substrates (_L_-Tyr and _L_-DOPA) [24,25].

As successfully shown by previous studies [26,27,28,29,30,31], MD simulations are an excellent technique to improve molecular docking results. Here, as described in the Materials and Methods section, five random replicas of 2 ns each were performed by using the Q program [32] version 6.0 [33] to provide ensembles for an improved binding process of 1,2,3-triazole-based KA into TYR. According to our MD results, all KA analogs are stable in the catalytic site of TYR*Bm* (Table 2), and the structural features of the TYR part are similar to the previous computational studies [19]. The root-mean-square deviation (RMSD) values of protein and ligand atoms are summarized in Table 2. As can be observed, these values range from 0.39 ± 0.04 Å (TYR-**6o** system) to 0.50 ± 0.04 Å (TYR-**6b**) for the protein part and from 0.47 ± 0.12 Å (TYR-**6j** system) to 0.81 ± 0.24 Å (TYR-**6f** system) for the ligand part. These results suggest a suitable stabilization of all KA analogs into the TYR catalytic site in all simulated complexes. Interestingly, all inhibitors maintained the interaction between the O atom of the carbonyl group of the KA part and Cu^2+^ ion (Cu2(B)) present in the TYR catalytic site, as observed for natural substrates (_L_-Tyr and _L_-DOPA) and the KA inhibitor [14]. Moreover, as previously evaluated [19], the CuDum model [34] applied for the description of Cu^2+^ ions appropriately described all important structural features. An RMSD plot of the protein and ligands with respect to the first snapshot of each system is provided as Appendix A. 

### 2.2. Binding Free Energy and Per-Residual Analysis

Due to the protein flexibility being not explicitly considered during the docking calculations, the prediction of accurate binding free energies can be difficult, and in this case, MD simulations can overcome it [35]. However, a suitable ensemble obtained from MD simulations should be evaluated to better describe the binding affinity of protein–inhibitor during molecular recognition. Combined with this feature, accurate and efficient methods to compute binding free energy (ΔG_bind_) are essential in computer-aided drug design [36]. Among these free energy methods, Linear Interaction Energy (LIE) [37] was selected for the prediction of experimental (∆G_EXP_) binding free energies of KA analogs complexed into TYR*Bm*. This approach has been applied successfully for TYR systems [18,19].

In Table 3, it can be observed all ligand-surrounding energies for KA analogs computed from MD simulations. Here, a total of 10 ns of MD simulations for each TYR system was chosen to compute LIE free energy (ΔG_LIE_) values. The empirical parameters α and β were chosen directly from the literature [38], see Appendix A for details. Particularly, the optimized value of γ (equal to 17.33) of LIE equation (Equation (1)) was calculated from the linear fitting with ∆G_EXP_ in order to include the Jahn–Teller effect included in the CuDum model for the bound models [34]. Particularly, it was found that excluding **6d**, **6g** and **6k** inhibitors resulted in a significantly better correlation with the ∆G_EXP_, so in all LIE discussions presented below, these KA analogs were excluded from the analysis, and a similar strategy was used by Carlsson et al. [27] and Vanga et al. [39].

The absolute binding free values for the 1,2,3-triazole-based KA inhibitors in the experimental data set are quite well-reproduced by the LIE approach (*r*^2^ = 0.97) (Figure 3). As all KA analogs are more potent TYR than KA inhibitors, our discussion is focused on considering the weakest (**6h**) and strongest (**6o**) TYR inhibitors, suggesting key features that explain their binding differences. Particularly, the ∆G_LIE_ value for **6h** is about 0.47 Kcal/mol higher than its experimental data (−7.14 Kcal/mol), while the ∆G_LIE_ value for **6o** is about 0.42 Kcal/mol lower than its experimental value (−9.91 Kcal/mol). As suggested by Ashooriha et al. [39], the good TYR activity shown by these KA analogs is explained by the presence of a 1,2,3-triazole ring. Therefore, new interactions found in that part of TYR inhibitors can provide insights about their inhibitory action. Therefore, to elucidate the energetic contributions of amino acid residues around the catalytic site of TYR, a residual decomposition analysis was carried out considering both van der Waals (*vdW*) and electrostatic (*ele*) contributions from the LIE equation.

The average *vdW* and *ele* interaction energies, overall KA analogs, for the TYR amino acid residues that contribute significantly to the ∆G_LIE_ are shown in Figure 4. As observed, in general, the *ele* contributions from TYR residues do not differ significantly for **6h** and **6o** systems (Figure 4B). Particularly, the interaction with Cu^2+^(B) is the most important contribution to the binding of KA analogs. Furthermore, the *vdW* contributions change significantly from **6h** to **6o** inhibitors (Figure 4A). The most evident difference can be observed for Phe197, Pro201, Arg209, Met215 and Val218, where the values change about 0.79, −0.54, −0.48, −0.56 and −1.48 Kcal/mol from **6h** to **6o** inhibitors, respectively (Figure 4). Interestingly, the interaction found between KA analogs and Arg209 occurs through π–cation stacking contact with the amino acid sidechain and 1,2,3-triazole ring of the KA analog (Figure 5). In the TYR–KA system, this residue interacts by an H bond with the carboxylic group of KA [14]. Finally, the **6o** inhibitor shows a strong *vdW* interaction with the Cu^2+^(B) ion of TYR, about 7.37 Kcal/mol lower than the **6h** inhibitor (Figure 4A).

## 3. Materials and Methods 

### 3.1. System Setup for Molecular Docking and MD Simulations

Initially, the 2D structures of KA analogs studied (Figure 6) were built into MARVINSKETCH (v. 22.18) program [40] and then optimized at the PM6 level [41] using GAUSSIAN09 [42] package.

The 3D structure of TYR from *Bacillus megaterium* (TYR*Bm*) with KA bound into the enzyme catalytic site was extracted from the Protein Data Bank (PDB code 5I38 [14]) as previously performed. The molecular docking calculations were carried out into Molegro Virtual Docker (MVD) version 5.5 [43], which has been applied successfully for TYR systems [18,19,20,21]. Particularly, for docking procedures, Cu^2+^ ions were included as van der Waals spheres at the catalytic site of TYR*Bm*. Our group has applied the MVD program successfully to describe the binding mode of TYR inhibitors [18,19,21]. Therefore, the same computational procedures were used for TYR-1,2,3-triazole-based KA systems. The MOLDOCK equations are detailed elsewhere [22].

For MD simulations, the best-ranked conformations of KA analogs were selected as starting points. The OPLS-AA [44] and TIP3P [45] force fields were used as parameters set to solute (TYR amino acids and KA analogs) and solvent subsystems, respectively. The OPLSA-AA parameters for KA analogs were computed by using the MACROMODEL package [46]. Particularly, a set of classical parameters proposed by Liao et al. [34], named the Cu^2+^ dummy model (CuDum), was used to describe the metal center of TYR*Bm*. The PROPKA approach [47] was used to set pKa values of all ionizable amino acid (AA) residues at neutral pH.

Each TYR-1,2,3-triazole-based KA system was solvated by a 20 Å radius simulation sphere of the TIP3P molecules [45] centered into the center of mass of the respective KA analog. The surface-constrained all-atom solvent (SCAAS) method [48] was used for polarization and radial constraints at the simulation sphere surface. All ionizable AA residues close to the sphere boundary were neutralized to account for dielectric screening [32]. A 10 Å cutoff was applied for computing non-bonded interaction energies, excluding only the atoms of KA analogs. The long-range electrostatic interactions were calculated using the local reaction field (LRF) multiple expansion method [49]. All atoms outside of the 20 Å radius simulation sphere were frozen to reduce computational costs [32]. The SHAKE algorithm [50] was used for solvent hydrogen bonds. 

The MD equilibration and production procedures for the bound (enzyme) and free (water) states are detailed in our previous study [19] using the Q6 program [50]. Each equilibrated system was submitted to a total of 10 ns of MD simulations from 5 randomized replicas of 2 ns each, where a time step of 1 fs was used and no positional restraints were applied. Particularly, for the free state, a weak harmonic restraint was used to maintain KA analogs in the respective center of their water simulation sphere. 

### 3.2. Binding Free Energy Calculations: LIE Method

A total of 10 ns of MD simulations from the production stage was used for binding free energy calculations according to the Linear Interaction Energy (LIE) method [37]. It uses the ensembles of the bound and free states of a respective ligand to compute their free energy difference [37]. The binding free energy ($\Delta G_{\mathrm{LIE}}$) value of each TYR*Bm* system was computed using the LIE equation (Equation (1)):(1)$$\Delta G_{LIE}=\alpha \left(\langle U_{vdW}\rangle {}_{bound}-\langle U_{vdW}\rangle {}_{free}\right)+\beta \left(\langle U_{ele}\rangle {}_{bound}-\langle U_{ele}\rangle {}_{free}\right)+\gamma$$

The α and β parameters are empirically scaling for the non-polar ($U_{vdW}$) and the polar ($U_{ele}$) terms, which are dependent on the chemical nature of the ligand [38]. These parameters can be obtained from the previous studies (α = 0.181 and β = 0.33−0.50) [38] or by linear fitting using experimental binding free energies ($\Delta G_{EXP}$) (Equation (2)). The average, 〈 〉, of van der Waals (“*vdW”*) and electrostatic (“*ele*”) interactions of “bound” and “free” states are computed using ensembles from MD production.
(2)$$\Delta G_{EXP}=\mathrm{RT}\ln IC_{50}+c$$
where the assay-specific constant (c) depends on the substrate concentration and the Michaelis–Menten constant ($K_{m}$) [51]. As this constant value does not affect the relative free energies, it can be implicitly included in the optimized value of γ (Equation (1)).

## 4. Conclusions

The present study investigated the accuracy of molecular docking and MD simulations in combination with the LIE method on a set of 1,2,3-triazole-based KA analogs, synthesized by click chemistry reactions as potent inhibitors of TYR enzyme. The TYR–inhibitor interactions were analyzed in detail, and it was found that the binding affinities of the selected KA analogs are driven mainly by van der Waals interactions. Particularly, a new π–cation stacking contact occurs between the Arg209 sidechain and 1,2,3-triazole ring of all KA analogs and may be related to their improved TYR activity. Our LIE analysis also agrees very well with experiments on TYR enzyme, providing a useful strategy for obtaining more potent TYR inhibitors and accurate predictions of TYR–inhibitor binding free energies.

## Figures and Tables

**Figure 1 molecules-27-08141-f001:**
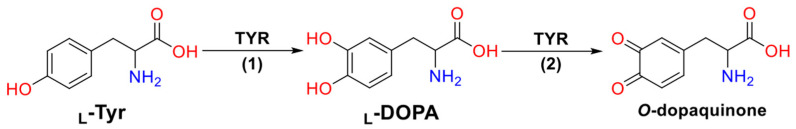
Reactions catalyzed by the TYR enzyme in melanogenesis.

**Figure 2 molecules-27-08141-f002:**
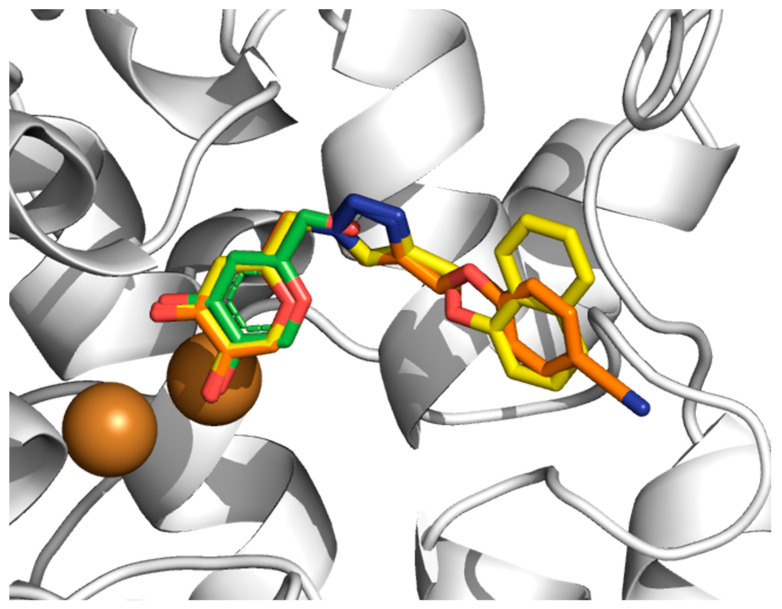
Three-dimensional overlapping of KA (green), **6h** (orange) and **6o** (yellow) into TYR*Bm* catalytic site by molecular docking. Cu^2+^ ions are shown as brown spheres. H atoms are omitted for clarity. All PDB files are provided as Appendix A.

**Figure 3 molecules-27-08141-f003:**
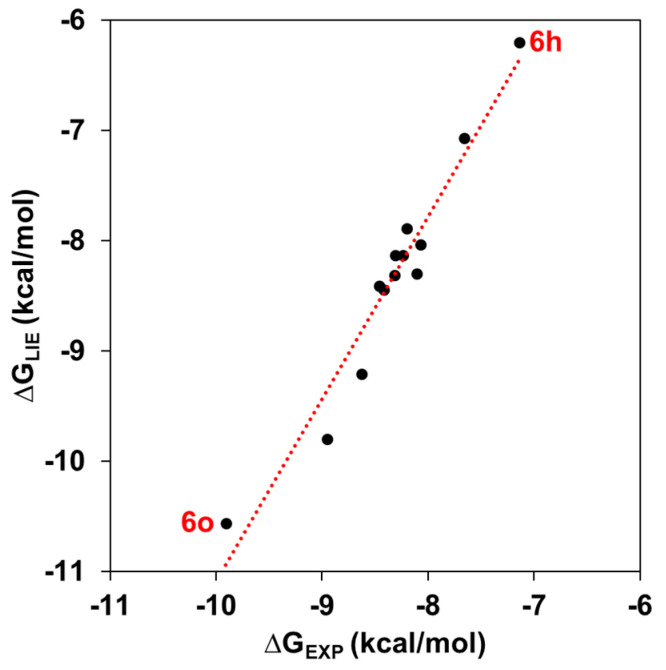
Calculated LIE (∆G_LIE_) vs. experimental (∆G_EXP_) binding free energies (Kcal/mol). The red dashed line represents the perfect agreement between ∆G_LIE_ and ∆G_EXP_.

**Figure 4 molecules-27-08141-f004:**
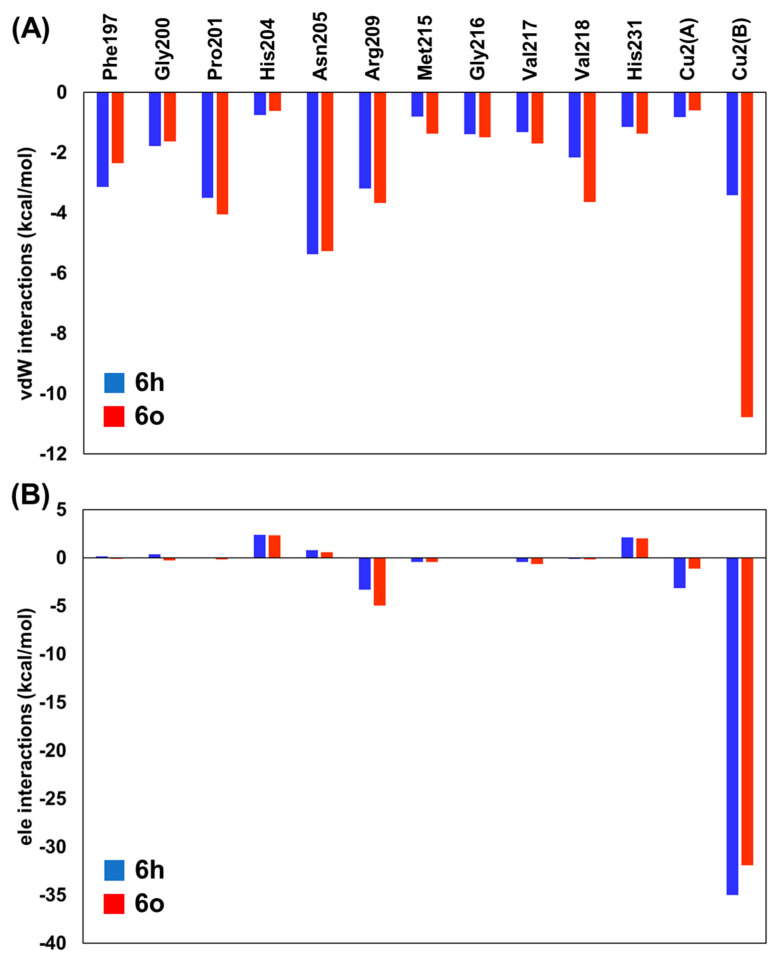
Residual interaction energies (in Kcal/mol) over **6h** (blue) and **6o** (orange) inhibitors used in the LIE calculations for the residues that contribute most to the ligand-surrounding (**A**) van der Waals (*vdW*) and (**B**) electrostatic (*ele*) contributions. The values for all TYR systems are provided as Appendix A.

**Figure 5 molecules-27-08141-f005:**
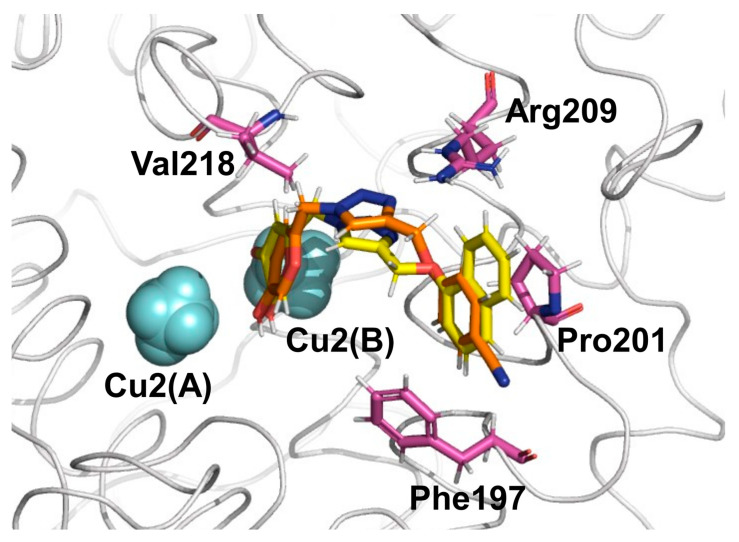
Four TYR residues make the most intense *vdW* contributions to binding for the inhibitors **6h** (orange color for C atoms) and **6o** (yellow color for C atoms) inhibitors. Cu^2+^ ions are shown as light blue spheres.

**Figure 6 molecules-27-08141-f006:**
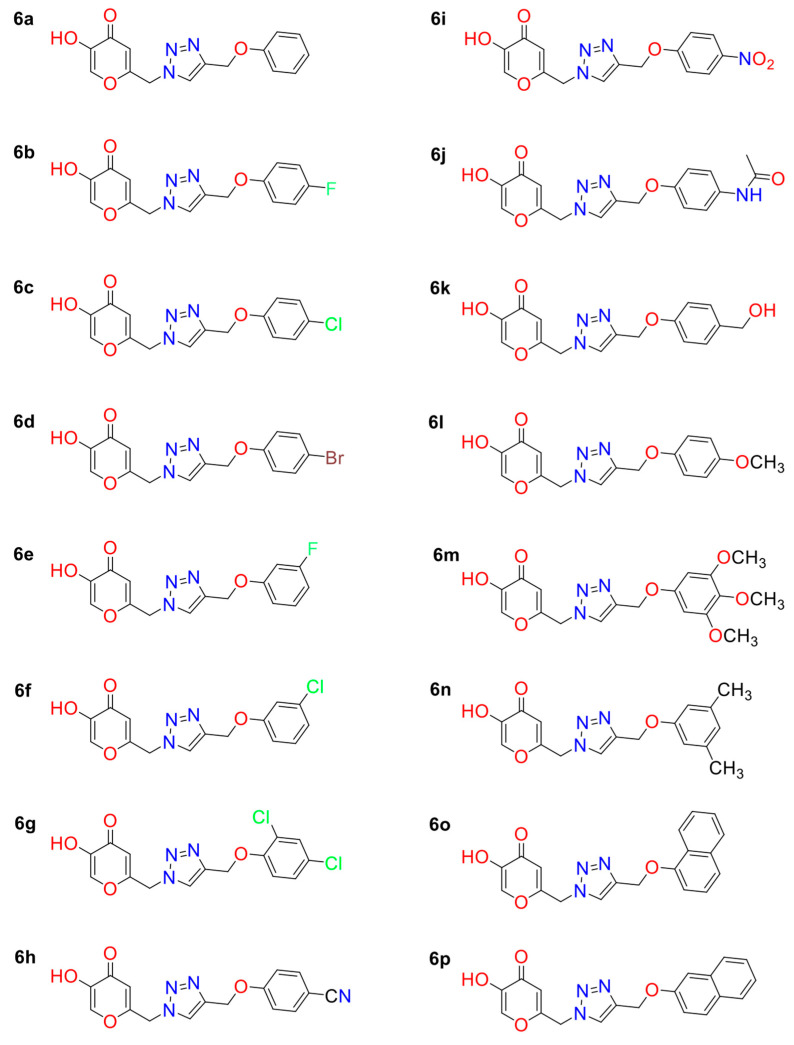
Two-dimensional structures of 1,2,3-triazole-based KA analogs.

**Table 1 molecules-27-08141-t001:** Molecular docking results (MOLDOCK scoring, Kcal/mol) and experimental activity (IC_50_, µM) of KA analogs in complex with TYR*Bm*.

KA Analog	MOLDOCK Scoring	IC_50_ *
6a	−120.78	1.33
6b	−132.33	0.88
6c	−132.03	0.69
6d	−128.15	6.80
6e	−125.68	1.07
6f	−136.38	0.99
6g	−129.55	1.12
6h	−139.06	6.29
6i	−132.92	0.52
6j	−135.60	2.64
6k	−132.85	1.32
6l	−125.93	1.24
6m	−130.46	0.87
6n	−130.17	0.74
6o	−130.15	0.06
6p	−131.51	0.30

* Values obtained from Ashooriha et al. [17].

**Table 2 molecules-27-08141-t002:** RMSD values (in Å) for protein and ligand atoms from TYR systems.

System	Protein RMSD	Ligand RMSD
TYR-6a	0.44 ± 0.05	0.47 ± 0.15
TYR-6b	0.50 ± 0.04	0.80 ± 0.20
TYR-6c	0.46 ± 0.07	0.53 ± 0.14
TYR-6d	0.40 ± 0.03	0.51 ± 0.16
TYR-6e	0.40 ± 0.05	0.60 ± 0.20
TYR-6f	0.44 ± 0.04	0.81 ± 0.24
TYR-6g	0.45 ± 0.03	0.79 ± 0.21
TYR-6h	0.43 ± 0.04	0.54 ± 0.13
TYR-6i	0.44 ± 0.05	0.51 ± 0.13
TYR-6j	0.43 ± 0.03	0.47 ± 0.12
TYR-6k	0.44 ± 0.04	0.62 ± 0.16
TYR-6l	0.43 ± 0.05	0.57 ± 0.13
TYR-6m	0.49 ± 0.06	0.55 ± 0.13
TYR-6n	0.47 ± 0.04	0.79 ± 0.19
TYR-6o	0.39 ± 0.04	0.49 ± 0.13
TYR-6p	0.46 ± 0.06	0.50 ± 0.13

**Table 3 molecules-27-08141-t003:** LIE-calculated (∆G_LIE_) and experimental (∆G_EXP_) binding free energies of KA analogs in complex with TYR*Bm*. All values are reported in Kcal/mol.

KA Analog	$\langle U_{vdW}\rangle {}_{free}$	$\langle U_{ele}\rangle {}_{free}$	$\langle U_{vdW}\rangle {}_{bound}$	$\langle U_{ele}\rangle {}_{bound}$	∆G_LIE_	∆G_EXP_
6a	−26.21 ± 0.01	−26.70 ± 0.49	−49.16 ± 0.98	−84.09 ± 0.24	−8.03 ± 0.45	−8.07
6b	−26.31 ± 0.06	−26.91 ± 0.10	−46.90 ± 0.34	−85.71 ± 0.63	−8.13 ± 0.34	−8.31
6c	−27.77 ± 0.02	−25.57 ± 0.17	−49.79 ± 0.41	−84.45 ± 0.70	−8.41 ± 0.40	−8.46
6e	−26.24 ± 0.03	−26.97 ± 0.31	−46.53 ± 0.21	−85.27 ± 0.76	−7.89 ± 0.44	−8.20
6f	−27.61 ± 0.02	−25.32 ± 0.18	−50.79 ± 0.90	−82.86 ± 0.89	−8.13 ± 0.56	−8.24
6h	−28.48 ± 0.08	−30.39 ± 0.45	−51.28 ± 0.29	−89.28 ± 0.78	−6.20 ± 0.47	−7.14
6i	−29.42 ± 0.04	−29.93 ± 0.02	−55.29 ± 0.51	−89.10 ± 0.26	−9.21 ± 0.20	−8.63
6j	−30.14 ± 0.08	−42.95 ± 0.30	−55.83 ± 0.37	−99.47 ± 0.97	−7.07 ± 0.53	−7.66
6l	−28.59 ± 0.03	−28.16 ± 0.16	−50.52 ± 0.68	−86.78 ± 1.01	−8.30 ± 0.56	−8.11
6m	−31.90 ± 0.10	−33.77 ± 0.61	−57.72 ± 0.40	−90.52 ± 0.99	−8.31 ± 0.68	−8.32
6n	−28.72 ± 0.09	−29.18 ± 0.74	−49.41 ± 0.79	−88.77 ± 0.90	−8.44 ± 0.76	−8.42
6o	−29.99 ± 0.30	−27.33 ± 0.05	−61.99 ± 0.37	−87.15 ± 0.24	−10.56 ± 0.23	−9.91
6p	−30.01 ± 0.20	−27.97 ± 0.43	−61.40 ± 0.91	−86.21 ± 0.22	−9.80 ± 0.44	−8.95

## Data Availability

Not applicable.

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
