# Peer review of "Computational Analysis of Triazole-Based Kojic Acid Analogs as Tyrosinase Inhibitors by Molecular Dynamics and Free Energy Calculations"

_molecules, 2022, doi:10.3390/molecules27238141_

Round 1
Reviewer 1 Report
This study are used the free energy for a set of 1,2,3-triazole- 13 based KA analogs as potent inhibitors of Tyrosinase (TYR) to find a new excellent inhibitor.Initially, molecular docking calculations predicted satisfactorily the binding mode of eval- 15 uated KA analogs, where the KA part overlays the crystal conformation of the KA inhibitor into the 16 catalytic site of TYR. The work was dull and crude. And can not be accepted in this journal.
Q1 What is the meaning of MD simulations ? I do see in this work ?
Q2 Is all the compounds were screened by others experiments? What did the authors do in this work ?
Q3 How long have you been doing MD? Were the system stable?
Reviewer 2 Report
The manuscript provide a comprehensive computational study using molecular docking, MD, and LIE approaches to study the various KA analogs as new and potent TYR inhibitors. Below are some issues need to be addressed:
1. (page 2)In Introduction, the author mentioned that Ashooriha et al synthesized a set of KA analogs. What are the advantages of these KA analogs over KA? If the KA analogs possess less side effects (e.g. lower toxicity), it should be mentioned in the introduction, as one of the motivations of the computational analysis of the KA analogs.
2. (page 2) Though the correlation between MOLDOCK and IC50 is low, can author add a scatter plot (x: MOLDOCK and y: IC50 ) to show the relationship between MOLDOCK and IC50?
3. (page 2) The author mentioned "Among the most important interactions, we can highlight the interaction with a key residue Arg209 through π–cation stacking interaction with 1,2,3-triazole ring present in all KA analogs." Any further explanation (of why Arg209 is special) or citation should be added here?
4. (page 4) Table 2 listed the RMSD values for protein and ligand atoms. Should the mean values be added as well?
5. (page 5) "lowest than" and "highest than" should be "lower than" and "higher than"
6. (page 5) The caption of Figure 3 is wrong (same as Figure 4). While Figure 4 shows the scatter plot of LIE-calculated (ΔGLIE) and experimental (ΔGEXP) binding free energies, can the author modify this figure: add another dash line representing ΔGLIE = ΔGEXP, and add error bar (available in Table 3) for each ΔGLIE value?
7. (page 6) Can author add color legend on Figure 4 to indicate 6h (blue) and 6o (orange) inhibitors?
Round 2
Reviewer 1 Report
The study can be aceepted at this version.